# Stress-Related Hormonal and Psychological Changes to Simulated and Official Judo Black Belt Examination in Older Tori and Adult Uke: An Exploratory Observational Study

**DOI:** 10.3390/sports12110310

**Published:** 2024-11-14

**Authors:** Simone Ciaccioni, Francesca Martusciello, Andrea Di Credico, Flavia Guidotti, Daniele Conte, Federico Palumbo, Laura Capranica, Angela Di Baldassarre

**Affiliations:** 1Department of Wellbeing, Nutrition and Sport, Faculty of Human Sciences, Education and Sport, Pegaso Telematic University, 80143 Naples, Italy; simoneciaccioni@yahoo.it; 2Department of Movement, Human and Health Sciences, University of Rome “Foro Italico”, 00135 Rome, Italy; francesca.martusciello@uniroma4.it (F.M.); federico.palumbo90@gmail.com (F.P.);; 3Department of Medicine and Aging Sciences, University “G. d’Annunzio” Chieti-Pescara, 66100 Chieti, Italy; andrea.dicredico@unich.it (A.D.C.); angela.dibaldassarre@unich.it (A.D.B.); 4Department of Human Sciences and Promotion of the Quality of Life, “San Raffaele” Open University of Rome, 00166 Rome, Italy

**Keywords:** combat sport, saliva, hormones, emotion, aging

## Abstract

This study investigated the psycho-physiological impact of a black belt examination. Older brown-belt judoka (Tori, F = 2, M = 4; age = 75.6 ± 4.5 yrs) and their 2nd–5th Dan black-belt coaches (Uke; M = 6; age = 36.5 ± 10.8 yr) were evaluated during a simulated and official examination and a resting day. Participants’ trait anxiety (STAI-Y2) was recorded prior to the study. State anxiety (STAI-Y1), ratings of perceived exertion (RPE), enjoyment (ENJ), and fear of falling (FoF) were collected 15 min before and after the experimental conditions. Saliva samplings at awakening (T0), PRE (T1), and POST (T2) exercise and during the recovery (15 min-T3, 30 min-T4, 60 min-T5) were collected for cortisol (sC), testosterone (sT), and alpha-amylase (sAA). Participants showed normal age-reference population trait anxiety. A difference (*p* ≤ 0.05) for role emerged for ENJ and sT only. For STAI-Y1, higher PRE values with respect to POST ones emerged (*p* = 0.005), and the highest values (*p* = 0.007) for PRE of the examination were with respect to the simulation. For sAA, differences for sampling were found in the examination conditions only, with peak values at T2 (370.3 ± 78.6 U/mL, *p* = 0.001). For sC, a significant peak value (0.51 ± 0.09 μg/dL, *p* = 0.012) emerged at T2 in the examination condition. With respect to Tori, Uke showed higher mean sT values in all conditions (*p* ≤ 0.05) and the highest T2 during examination (712.5 ± 57.2 pg/mL). Findings suggest the relevance of monitoring psycho-physiological stress-related responses in judo for optimizing both coaching effectiveness and sport performance, especially in older judo practitioners.

## 1. Introduction

Judo is a lifelong educational martial art and an Olympic combat sport since 1964, with competitions encompassing standing-up and ground fighting organized in relation to the age, sex, weight, and skill level (e.g., belt color) of the athletes [1,2]. Judo practitioners need to constantly adjust their physical efforts and technical-tactical performances according to their opponents [2]. In mastering sparring with an opponent (e.g., randori) and sequences of specific movements (kata), judo practitioners progress their skill levels from white-belt color (6-kyu) to brown-belt color (1-kyu) and black belt 1–10 Dan. To advance from one level to the next, an athlete must fulfill theoretical knowledge related to judo and practical sport-specific executions.

Simulated and official judo competitions have been considered suitable to evaluate the external load (time-motion and technical-tactical analysis), and the athletes’ psycho-physiological (questionnaires and salivary biomarkers) efforts [3,4,5]. In combat sport athletes, salivary α-amylase (sAA) has been used to evaluate rapid stress responses of the sympathetic changes in autonomic activity, whereas salivary-free cortisol (sC) and salivary testosterone (sT) evaluate the opposite role in the regulation of the hypothalamus–pituitary–adrenal (HPA) activity during training and competitions [6]. In addition, the sT/sC ratio is considered a sensitive marker to training volume and physiological stress [6]. With respect to a resting day, the psycho-physiological arousals in the morning of the competitive day could help isolate the responsiveness to competition pressures [6,7]. Although salivary markers show increases after a competition, a large inter-individual variability has been reported, probably due to a confluence of contextual, behavioral, and psycho-physiological factors [6,7].

Master athletes are older individuals who train and compete in tournaments, offering a barometer of age-related performance and physical and psychological changes, being considered examples of successful aging [8]. Through their sports training and competition, master athletes develop positive adaptations benefiting their healthy living [9,10,11,12]. Recently, judo has been proposed as a multimodal sport-based intervention for older practitioners to enhance their functional fitness, walking performances, physiological and psychological well-being, social interactions, and quality of life, as well as to prevent fall-related injuries and reduce stress levels, negative feelings, and depression [13,14,15,16,17,18,19]. However, it is important that coaches adapt technical elements favoring the effective development of self-defense, self-control, and discipline of older practitioners toward the progression from white to black belts [16,17,18,20,21]. During a judo belt examination, the examinees (Tori) undergo an oral test on judo theory and perform judo techniques with an experienced partner (Uke). Despite the fact that a progression toward the back belt could nurture self-satisfaction and accomplishment feelings in older judo practitioners, the official examination could elicit anxiety, which is characterized by feelings of tension and uneasiness, worried thoughts, physical changes, and negative moods for possible failure [3,22].

The State–Trait Anxiety Inventory (STAI) allows the investigation of anxiety levels, as it distinguishes between transient emotional states (state anxiety) and a stable predisposition to stress (trait anxiety) [23]. This psychological measure presents a high relevance in linking anxiety with physiological stress markers, such as cortisol, testosterone, and alpha-amylase. In combat sports, anxiety has been associated with negative performance in evaluative situations perceived as intimidating, uncontrollable, and unpredictable [3,22,24]. Although physical activity could reduce the age-related increases in anxiety, older individuals could experience increased stress-related anxiety during an evaluation due to their reduced physical, cognitive, or behavioral performances, unless the examinee is familiar with the context and the examiner [25,26]. However, there is a lack of information on stress-related psycho-physiological changes in older judo practitioners during training and under official judo belt examination.

Therefore, the present study aims to assess the psycho-physiological impact of a simulated and an official judo black-belt 1st Dan examination on older Tori and on their adult Uke. To address the gaps in the literature on age-related stress responses, the main hypotheses of the study relate to differences, if any, between older judoka and their younger more experienced counterparts in different settings (i.e., resting condition and simulated and official examinations), as follows:Due to evaluation-related pressure in a formal and demanding context, older judo-ka would experience higher stress levels during an official examination compared to simulated and resting conditions;Due to differences in experience and physical capacity between groups, older judoka would face higher stress during an official examination compared to skillful black belt adult Uke.

## 2. Materials and Methods

### 2.1. Study Design

The local Institutional Review Board (CAR48/2020/INT/2022) approved this exploratory observational study with the experimental group serving as their own control. The volunteers in the present study signed a written informed consent to participate in a 3-day data collection encompassing a simulated and an official judo black belt examination and a resting day. Anonymity was guaranteed by coding the personal information of the participants.

To evaluate the participants’ stress responses during simulated and official judo black belt examination, the experimental design included concomitant measurement of psychological and physiological parameters [3].

### 2.2. Participants

The experimental sample included a Tori who performed the judo techniques and their partner Uke. To account for the potential physical differences associated with aging, participants’ health status, fitness levels, and previous judo experience were also considered. Therefore, to be eligible for the present study, Tori participants had to be (i) healthy individuals who have been officially certified their health condition suitable for sport practice following a thorough medical check by a specialized sports medicine physician; (ii) older adults, aged ≥65 years old; (iii) without injury conditions potentially affecting their exam participation; and iv) available to voluntarily participate in the study. Regarding Uke, the inclusion criteria encompassed (i) being a certified judo coach; (ii) having experience in coaching older judo practitioners; and (iii) being available to volunteer for the study.

Both Tori and Uke participants answered the American Alliance for Health, Physical Education, Recreation, and Dance (AAHPERD) questionnaire, which consists of the activity and health history, including physical activity level, weight and height, recent weight loss/gain, recent illnesses, recent hospitalizations during last 5 years, tobacco smoking, alcohol consumption, health concerns, medications, and/or dietary supplements [27]. Furthermore, questions regarding educational background and judo practice-related experience were asked. Participants were familiarized with the experimental procedures. Six older Tori brown belt participants undergoing a black belt examination (F = 2 M = 4; age = 75.6 ± 4.5 yr) and six adult (M = 6; age 36.5 ± 10.8 yr) black belt Uke participants (2nd–5th Dan) volunteered for this study. Starting from a white belt and no previous martial arts experience, for six years, the Tori engaged in two 1-h judo training sessions weekly, whereas the Uke had a judo experience of 15–30 years.

Older judo practitioners represent an extremely small number of their judo and older individual cohorts [8,18,28,29]. To guide the sample size of the present explorative study, a power analysis targeting a medium effect size (Cohen’s d = 0.5) with a power of 0.80 and an alpha level of 0.05 was conducted. Therefore, 12 individuals were determined as sufficient to detect meaningful within-subject changes in psycho-physiological markers.

### 2.3. Procedures

#### 2.3.1. Judo Black Belt Examination

According to the Italian Judo Wrestling Karate and Martial Arts Federation (FIJLKAM), the white belt is conferred to all novice judo practitioners, whereas their skill progress from yellow to brown belt is verified by internal or external evaluators in the judo club where the practitioners habitually train. Conversely, to obtain their 1st Dan, judoka need to pass a regional-based examination chaired by the regional Vice-President including coaches of a national list. To prepare for the examination, the Tori must train their skills (e.g., *taiso*-general exercise, *kata*-prearranged sequences, *randori*-“free practice”, *uchikomi*-repetition training). When the eligibility criteria of technical skills and theoretical competences are met, the participants apply to the Dan examination, providing to the National Judo Federation personal details, training history, and any necessary documentation.

Table 1 summarizes the official examination program performed by the older Tori. The theoretical knowledge examination encompassed judo theory, rules, terminology, and etiquette. To ensure safety for the older judoka during the practical examination, adapted techniques (e.g., kneeling *kata-guruma*) were allowed. In opposing their black belt Uke, the Tori were required to be proficient in executing standing throws (*tachi-waza*), groundwork (*ne-waza*), other techniques (*ukemi waza*-breakfall techniques, *taisabaki*-rotational movements, *ayumi*/*tsugi ashi*-linear movements), and three different kata: (i) *Nage-no-kata*, the first nine techniques of a prearranged series of movements that demonstrate a wide range of throwing techniques performed from both the right and left stances; (ii) *Ju-no-kata*, the first five techniques of a “choreography” focusing on the principles and movements of gentleness and flexibility; and (iii) *Seiryoku Zen’yo Kokumin Taiiku*, the individual or solo practice (*tandoku-renshu*) of shadow kicks and punches. The judo black belt official examination took place in a formal environment, adhering to the standard procedures set by the national judo federation, with a panel of official examiners present and strict evaluation criteria. In contrast, the simulated examination was conducted during a training camp, replicating key aspects of the official test, such as the execution of the same judo techniques under evaluation and the presence of high-level judoka (8th dan) judging the technical executions as well as asking judo-related theoretical questions. However, it differed in terms of the setting (training) and the absence of official examiners, reducing the external pressure typically associated with the formal official belt examinations. This design allowed for the assessment of stress responses in both high-stakes and controlled conditions.

#### 2.3.2. Measurements

##### Psychological Assessments

For the psychological parameters, the individual perceived anxiety was investigated by means of the 20-item State–Trait Anxiety (Form Y) Inventories encompassing (i) the State Anxiety Inventory administered before the simulated and the official judo black belt exam conditions to assess the practitioner’s momentary emotional state characterized by feelings of apprehension and tension and (ii) the Trait Anxiety Inventory, administered during a resting day to help to clarify the interpretation of the test-taking outcomes by assessing the relatively stable individual tendency to respond with anxiety to perceived threats [23]. To verify the potential impact of exercise intensity and enjoyment, the rating of perceived efforts (CR-10 RPE [30]), the Falls Efficacy Scale (FES [31]), and the Visual Analogue Scale (VAS)-Enjoyment [32] were administered during the simulated and official judo black belt examination conditions.

After being informed that there were no right or wrong responses, the participants individually completed the psychometric questionnaires, supervised and supported by an investigator, in case help was needed. The participants were required to consider the statement “how are you feeling right now” for the STAY-Y1 questions and “how you generally feel” for the STAI-Y2 ones [23]. Whilst the STAI-Y2 was administered during a resting day 25 days before the experimental period, the STAI-Y1 was administered 15 min before and after the experimental sessions. Individual STAI-1 and STAI-2 values were compared to age-reference norms for the Italian population and expressed in pt [33]. STAI-Y reliability levels (e.g., Cronbach’s alpha) range from 0.84 to 0.86 in the Italian older population [34]. At 30-min of the recovery phase, participants were asked to answer the questions: “How was your examination?” [35], “How did you experience your examination?” [36], and “Were you afraid to fall?” [37] to assess their ratings of perceived efforts, enjoyment, and fear of falls, respectively. The CR-10 RPE scale has demonstrated reliability coefficients of approximately 0.80 [38], while the FES exhibited Cronbach’s alpha values above 0.90 in previous studies [39]. Previously used in research on judo for older adults [32], the VAS-Enjoyment was found to be correlated with exercise level (r = 0.36) [36].

##### Saliva Sampling

For the psycho-physiological parameters, saliva samples were collected at awakening (T0), before (PRE, T1), and at the end (POST, T2) of the simulated and official judo black belt exam and at 15 (T3), 30 (T4), and 60 (T5) min of the recovery phase to assess stress-related sAA, sC, and sT responses. During a resting day (mean temperature = 25 °C, mean humidity = 45%), samples were collected indoors at the same times as the experimental days (simulated: mean temperature = 24 °C, mean humidity = 42%; official examination: mean temperature = 28 °C, mean humidity = 37%). The 6-time point measurements for the saliva sampling have been selected according to the literature of combat sports [40,41,42]. The awakening time was between 7:00 and 7:30 a.m., T1 was between 10:00 and 10:15 a.m., and T2 was between 10:25–10:40 a.m., followed by the 15 (T3), 30 (T4), and 60 (T5) min of the recovery phase.

Cotton swabs and collecting tubes (Salivette, Sarstedt, Germany) were used to obtain saliva samples (>0.05 μL). The participants were instructed to place the cotton swab into their mouths for 2 min and to chew 20 times, under the supervision of an investigator. The saliva-collecting tubes were centrifuged at 3000 rev/min for 15 min at 4 °C. Samplings were stored at −80 °C until they were assayed and tested in the same series and in duplicate. According to the manufacturer’s instruction (Salimetrics, LLC, Carlsbad, CA, USA), enzyme immunoassay kits (#1-3002, #1-2402) were used to measure sC and sT concentrations, and kinetic enzyme assay kits were used to measure sAA levels (#1-1902). Plate readings were performed at 1 and 3 min at 405 nm using Varioskan™ LUX multimode microplate reader (#VL0000D0, ThermoFisher Scientific, Waltham, MA, USA). The minimal concentrations of sT and sC that can be distinguished from 0 is 0.458 pg/mL and 0.007 μg/dL, respectively. Regarding the sAA kit, the lower limit of sensitivity is governed by the change in absorbance. Samples that yield values ≥ 2.0 U/mL (at a 1:200 dilution) result in a reliable value. The functional sensitivity of sC and sT samples tested as 20 replicates and 3 separate rounds is 0.018 μg/dL and 0.68 pg/mL, respectively.

### 2.4. Statistical Analysis

Data analyses were conducted (SPSS 25.0, Chicago, IL, USA) with a 0.05 level of significance. Descriptive statistics included means, standard deviations (SD), and standard error of the mean (SEM). Prior to the analysis, the Shapiro–Wilk test was applied to ascertain the normality of data distribution. Thus, the non-parametric Mann–Whitney U test was applied to verify differences between Tori and Uke and between conditions (Simulation, Examination) for RPE, FoF, and ENJ. Friedman’s test for repeated measures analysis was applied to salivary samplings (T0–T5) and conditions (rest, simulation, examination). When differences emerged, post hoc comparisons with Bonferroni corrections were used. The results were normally distributed; STAI-Y1 was analyzed by means of a 2 (role: Tori, Uke) × 3 (condition: rest, simulation, examination) × 2 (time: PRE, POST) ANOVA for repeated measures.

## 3. Results

The judo committee awarded the participants a 1st Dan. A difference (*p* ≤ 0.05) in role emerged for ENJ and sT only. For the other variables, group data were collapsed.

Trait and State anxiety (Table 2) resulted in normal results for the age-reference population expressed in T-scores (pt). For the role, no main effect was found. For STAI-Y1, differences emerged for time (*p* = 0.005) and the time × condition (*p* = 0.005) interaction. In general, higher PRE values with respect to POST ones emerged. Post hoc comparison for the time × condition interaction maintained only a difference (*p* = 0.007) for PRE, with higher values for examination with respect to simulation.

Table 3 shows the means and standard deviations of ENJ, RPE, and FoF recorded at the end of the simulated and examination conditions. No difference was found for FoF, with overall very low scores (1.6 ± 0.8 pt). For RPE, higher mean values emerged for Uke (simulation = 6.0 ± 2.4 pt; examination = 6.3 ± 2.7 pt) with respect to Tori (simulation = 4.5 ± 2.7 pt; examination = 2.7 ± 1.6 pt). However, no main effect was maintained after the Bonferroni correction. For ENJ, a difference was found for the role in both the simulation (*p* ≤ 0.004) and the examination (*p* ≤ 0.006) conditions, with Tori showing higher values with respect to Uke.

For sAA, Figure 1 reports the means and SEM, with differences for sampling (*p* = 0.002) found for the examination condition only, with the lowest values recorded at T0 (81.9 ± 11.9 U/mL) and the highest values recorded at T2 (370.3 ± 78.6 U/mL). For some conditions, a difference emerged for T2 between examination with respect to simulation (172.8 ± 38.9 U/mL) and rest (132.9 ± 33.4 U/mL).

For sC (Figure 2), differences emerged for sampling in the rest (*p* < 0.001), simulation (*p* = 0.001), and examination (*p* = 0.026) conditions, with the highest values recorded at T0 (rest = 0.58 ± 0.07 μg/dL; simulation = 0.50 ± 0.05 μg/dL; examination = 0.57 ± 0.10 μg/dL). The Bonferroni correction did not maintain a difference for examination. For some conditions, a difference (*p* = 0.012) was found only for T2 between the highest value recorded during the examination (0.51 ± 0.09 μg/dL) and the values of the simulation (0.21 ± 0.03 μg/dL) and rest (0.27 ± 0.03 μg/dL). For sampling, post hoc analysis maintained differences (*p* range: 0.007–0.045, SE range: 0.047–0.057 μg/dL) between the peak values recorded at awakening (T0: 0.55 ± 0.05 μg/dL) and the other sC samplings. Similar sC values were recorded at awakening (simulation = 0.496 ± 0.049 μg/dL; examination = 0.570 ± 0.099 μg/dL; rest = 0.580 ± 0.072 μg/dL).

For sT (Figure 3C), differences emerged between Uke and Tori for examination (T0: *p* = 0.015; T2: *p* = 0.015; T3: *p* = 0.015) and rest (T0: *p* = 0.015; T1: *p* = 0.026) only, with Uke showing higher mean values in all conditions with respect to those of the Tori. No difference was found for sampling. For conditions, Uke showed the highest T2 values for examination (712.5 ± 57.2 pg/mL), intermediate for rest (500.6 ± 40.3 pg/mL), and lowest for simulation (398.5 ± 33.7 pg/mL) conditions.

For sT/sC (Figure 4), no difference was found between groups. A difference between sampling emerged only for rest (*p* < 0.001) with the highest values at T2 and T4 (0.19 ± 0.02 AU).

## 4. Discussion

The novelty of this study is the investigation of the stress-related psychometric (ENJ, FoF, RPE, STAI-Y1) and psycho-physiological salivary (sC, sAA, sT) responses in simulated and official judo examination and a rest day of adult (range: 26–50 yrs) black belt Uke and older (range: 70–83 yrs) brown belt Tori. The main findings highlighted similar STAI-Y1, STAI-Y2, RPE, FoF, sC, sT/sC, and sAA values between the Tori and the Uke; whereas, differences between the two groups emerged for ENJ and sT only.

In sports, emotions are important components of performance. As expected, both Tori and Uke showed increases in their anxiety level prior to the simulated and the official examination, mirroring their psychological readiness yet not exceeding the normal values for their age-reference populations probably due to their conviction in their ability to perform effectively both the theoretical and practical parts (prearranged technique sequences, free demonstration) required to acquire a black belt. Despite the powerful influence of negative aging stereotypes on memory performance and movement proficiency, the older Tori’s habitual practice of judo likely bolstered their self-confidence in memory ability, reduced their fear of falling, and helped them manage anxiety during the challenging conditions of the official examination, as substantiated by the present simulation and examination enjoyment scores [16,17,37,43]. To note, the Tori started their judo program six years prior to the present study, and every year, they passed a belt examination, which could substantiate the theory-based mechanisms of the positive effects of regular gratification and increasing reward on volitional maintenance behaviors and further goal settings [44]. Considering that individuals aged 55+ yrs are the most sedentary European group, the Tori engagement in judo in later years proved their high exercise adherence associated with personal satisfaction, high self-determination, and improved identity, leading to the recognition of the national sport federation, which also underlies the importance of the social and institutional environment for the regulation of a positive self-value [13,14,17,18,19].

To our knowledge, this study provides novel information on the simultaneous performance of the black belt examinees and their judo coaches, who are central to the development of the athletes’ performances. Like athletes, coaches also encounter a variety of stressors, with responses determined by a perceived asymmetry between environmental pressures and the individual’s ability to handle resources [45]. Coaching older judo practitioners can be significantly stressful due to numerous tasks and duties, including the need for injury prevention [18,20,21,45]. Additionally, Uke’s responsibilities in organizing the residential camp for the simulated event and their crucial role in supporting Tori during the black belt examination likely contributed to their perceived effort. Finally, the Uke might have considered the judo belt examination of their Tori as a test for their coaching effectiveness, which might have determined their lower enjoyment and higher perceived efforts with respect to those of the actual examinees [46]

Exercise-related psycho-physiological effects during training, recovery, and competitions have been studied as markers of stress in young judoka [3,47]. The present saliva sampling protocol allowed us to observe temporal changes in sAA, sC, and sT during simulated and official examination conditions that are assumed to be accompanied by high physiological and psychological stressors. The novel information on stress-related biomarkers collected in the adult and older judoka contributes to the current stress-related studies of physiological alterations of HPA and the SAM axes during combat sport performances [3,4,5]. During the rest day, sAA exhibited the lowest values. The effects of aging on α-amylase secretion have not yet been clearly established, with some authors highlighting a constant sAA profile throughout adult life and others reporting age-related differences in diurnal patterns [48]. The lack of differences between the adult Uke and older Tori presenting around a mean difference of 40 years substantiates a constant sAA profile throughout adult life [48]. To note, comparable low sAA values were found in the rest and simulation conditions, even though the latter implied moderate physical efforts (e.g., 4–6 pt RPE scale) [4], which could induce physiological stress with sAA increases. Conversely, during the examination day, sAA started increasing from awakening at T1, reaching significant peak values at T2 despite the Tori-reported effort of light-to-moderate intensity (RPE range: 1–5 pt) and the Uke moderate-to-hard intensity (RPE range: 3–9 pt). Therefore, the sAA variations observed right after the examination likely substantiate a psychological stress response [4]. According to the literature, a fast recovery followed the significant peak increases at the end of the stressful condition, reflecting changes in sympathetic activity [48].

According to its circadian rhythm, the present findings showed similar sC values for Tori and Uke. The highest sC levels were found at awakening (T0) and significant decreases were observed during the rest and the simulation days, which mirrored the diurnal cortisol secretion pattern represented by a sharp awakening response and a progressive decreasing slope until the end of the day [49]. The age-related contribution to the sC levels is controversial, with Haney et al. [50] highlighting no relevant variation in cortisol as a function of age and Piazza et al. [51] showing high levels of cortisol in saliva in older individuals. The situation is even more complex when considering the altered sleep-wake cycles with advancing age that could potentially influence the sC profiles [52]. Also, a link between sC and positive or negative feelings is not fully established [53]. In the literature, significant sC increases post-competition compared to pre-competition in young athletes have been reported [3,4]. In the present study, a significant increase was also observed for sC after the examination (T2), substantiating a coordinated SAM and HPA reaction to generate a stress response even though the post-examination sC recovery resulted more slowly compared to sAA [4].

According to the age-related reduction in the number and secretory capacity of the Leydig cells, lower sT values were observed in older Tori with respect to their adult Uke counterparts, with significant differences at awakening and T1 in the rest condition. Whilst the anticipatory response of cortisol has long been recognized, the anticipatory response of testosterone could not be considered reliable, with an inversely proportional trend between testosterone and cortisol reported in stress situations of a noncompetitive nature. For the Uke, the highest sT levels found at awakening and after the official examination substantiate the stress-related responses to challenging judo situations [3]. Although the sT:sC ratio has been used as an exercise stress index [4], it could be speculated that the lack of differences between Uke and Tori is attributable to the significantly higher sC values observed in both groups.

In the present study, the low values of all the considered salivary markers recorded during the simulation condition deserve some consideration. In tapering for the examination, the simulation condition had been organized in a dojo located in the countryside as part of a relaxed intergenerational judo social event aimed to foster the older practitioner’s wellbeing and memory function and to balance positive–negative emotions in achieving a black belt [16,54,55]. In line with the literature on stress-related effects of tapering prior to a youth judo competition, older practitioners also showed low RPE values and high enjoyment, suggesting a strong relationship to their low stress, positive emotional state mirroring intrinsic motivation in performance, and ability to cope well with the inherent psycho-physiological stress from the examination [56]. Furthermore, these data substantiate previous studies, showing a down-regulation of physiological arousal in older individuals due to positive affective states being higher with respect to what is typical for them [53].

## 5. Limitations, Practical Implications, and Future Directions

This study examined the simultaneous execution of judo techniques by older Tori and adult Uke during a judo black belt examination, highlighting distinct psycho-physiological stress responses between the two roles. The findings revealed that older Tori exhibited limited stress-related changes compared to their Uke. This suggests that older Tori may perceive the black belt examination as a personal milestone, while their coaches acting as Uke experience greater stress due to the responsibility of ensuring proper execution and guidance. These differential stress responses underscore the complex dynamics between performance and mentorship in judo. One of the limitations of the study is the relatively small sample size, which may reduce the generalizability of the findings to the broader judo population. Additionally, the study’s simulated condition may not fully mirror the stressors present in actual judo examinations, potentially affecting the ecological validity of the results. The findings have important practical implications for judo training and coaching, particularly in the context of black belt examinations. Coaches and instructors should recognize that the stress experienced by Uke, especially when they act as mentors or evaluators, may differ significantly from that of Tori. Tailoring training and psychological preparation to account for these role-based stress differences could improve performance outcomes and enhance the overall experience of both practitioners. Additionally, acknowledging the lower stress levels in older Tori may encourage broader participation among older judo practitioners, supporting their physical and psychological well-being. To further explore the psycho-physiological stress responses in judo, future studies should include larger and heterogeneous samples, including practitioners from different age groups, skill levels, and backgrounds. This would enhance the external validity of the findings. Additionally, incorporating a variety of psycho-biological assessments (e.g., cathepsin and salivary catecholamines levels, personal achievement goals) would provide a more detailed analysis of stress-related responses. Future research should also aim to increase the ecological validity of the study by simulating real-world examination conditions, including assessments from official judo judges and considering competitive envi-ronments. Such approaches would offer a more realistic and nuanced understanding of stress dynamics in judo examinations.

## 6. Conclusions

This study underscores the importance of investigating stress-related responses in judo, particularly among older practitioners and their coaches during performance assessments. The findings highlight the need for further research into the psycho-physiological demands of judo examinations, focusing on how these stressors affect both the well-being of older judoka and the coaching dynamics. This work encourages the scientific community to expand the understanding of age-related factors in judo, particularly regarding stress management and the balance between personal achievement and coaching responsibilities. In this context, the present study represents the first attempt to evaluate stress-related hormone concentrations during a judo black belt examination, exploring how practitioners manage the psychological and physiological demands of exam-based settings. Further research is essential to clarify the specific mechanisms underlying the stress responses observed in this study, which indicate that in preparing for a judo black belt examination, it is crucial to incorporate simulation conditions that allow anxiety state management, balance positive and negative emotions, and promote enjoyment. Notably, the higher stress-related responses observed in Uke, compared to Tori, suggest that judo coaches should be mindful of their own well-being and emotional regulation while supporting their athletes. Monitoring these factors could play a key role in optimizing both coaching effectiveness and athlete performance.

## Figures and Tables

**Figure 1 sports-12-00310-f001:**
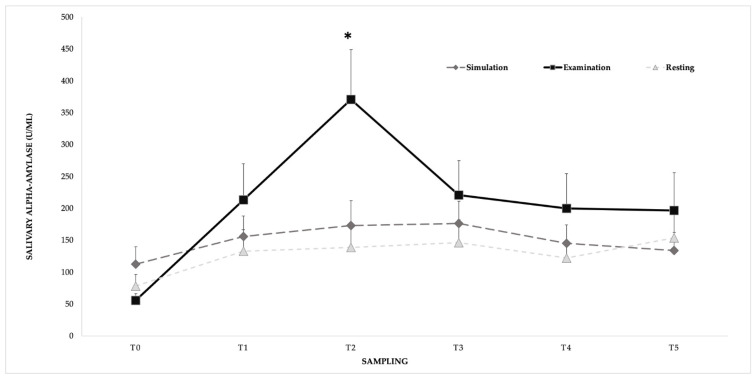
Means and standard errors of the mean (SEMs) of salivary alpha-amylase recorded at awakening (T0), before (T1) and after (T2) the exercise, and during the recovery (15 min-T3, 30 min-T4, 60 min-T5) phase of a simulated and official Judo black belt examination, and on a rest day. Note: * *p* = 0.001 T2 examination with respect to the other salivary alpha-amylase samplings.

**Figure 2 sports-12-00310-f002:**
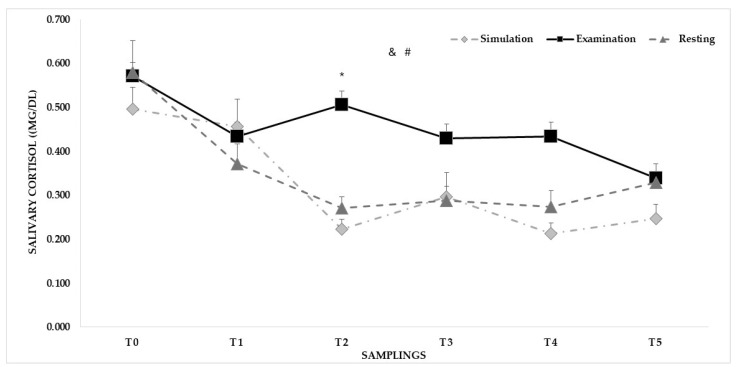
Means and standard errors of the mean (SEMs) of salivary cortisol recorded at awakening (T0), before (T1) and after (T2) the exercise, and during the recovery (15 min-T3, 30 min-T4, 60 min-T5) phase of a simulated and official judo black belt examination, and on a rest day. Notes: & and # indicate differences between samplings for the resting (*p* < 0.001) and simulation (*p* = 0.001) conditions, respectively. * indicates a difference (*p* = 0.012) between conditions at T2.

**Figure 3 sports-12-00310-f003:**
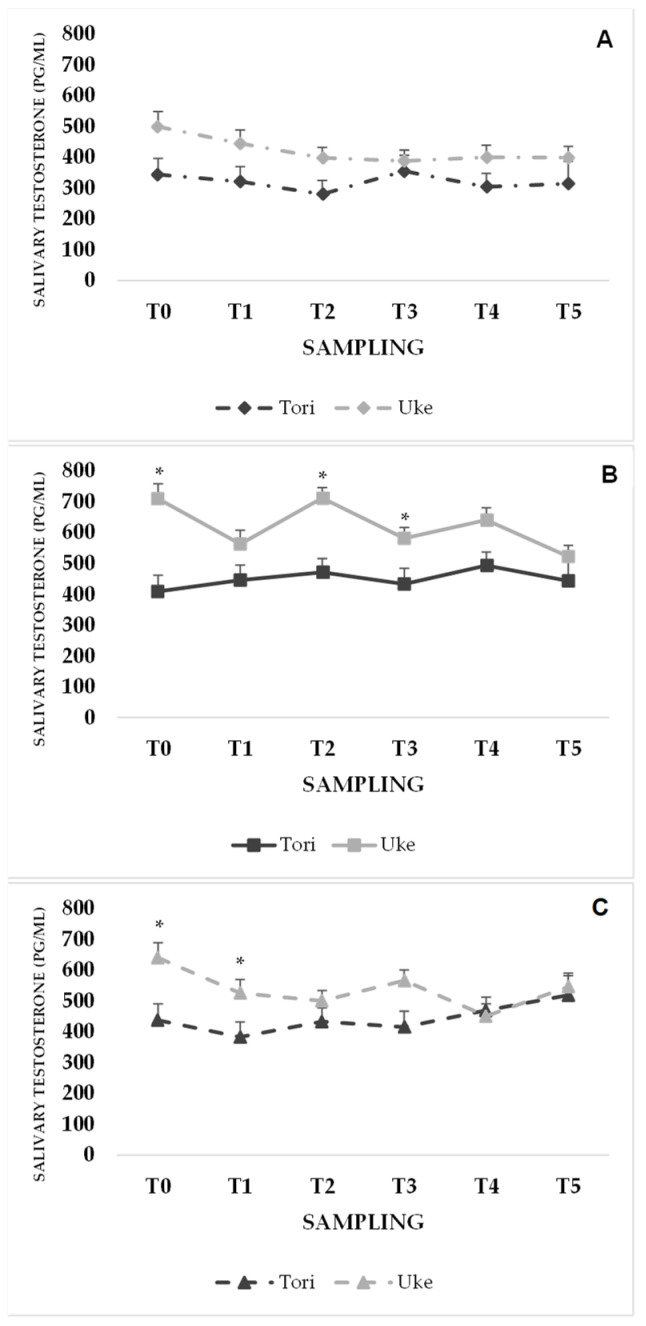
Means and standard errors of the mean (SEMs) of sT for Tori and Uke across different time points. Note: (**A**) simulation, (**B**) examination, (**C**) rest. * indicates *p* < 0.05 between groups.

**Figure 4 sports-12-00310-f004:**
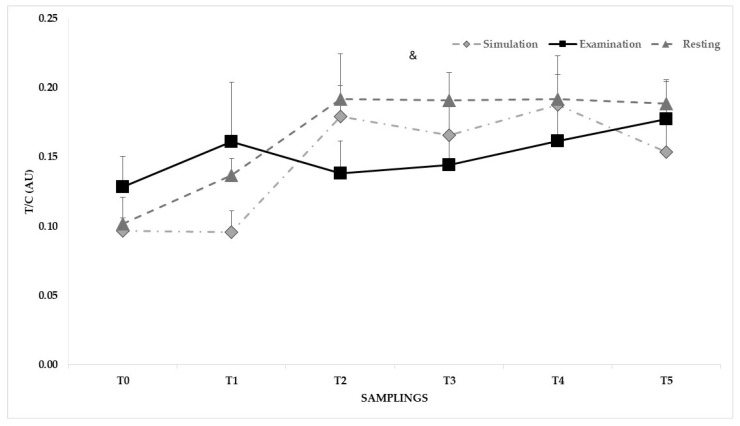
Means and standard errors of the mean (SEMs) of T/C at awakening (T0), before (T1) and after (T2) the exercise, during the recovery (15 min-T3, 30 min-T4, 60 min-T5) phase of a simulated and official Judo black belt examination, and on a rest day. Notes: & indicates a difference (*p* < 0.001) between samplings for the rest condition.

**Table 1 sports-12-00310-t001:** 2022 FIJLKAM * program for the 1st Dan graduation.

Theory											
Philosophy and history of judo			Federal organization			Refereeing: terms and gestures			Techniques names		
**Practice**											
*Fundamentals*											
*Shisei*-Positions			*Kumi kata*-Grips			(In)balance methods-*Kuzushi*			Moves-*shintai* and *tai-sabaki*		
*Breakfall techniques-Ukemi waza*											
*Ushiro*-Backwards				*Mae*-Forwards				*Yoko*-Lateral			
*Nage-no-kata*											
*Te-waza*-Hand techniques				*Koshi-waza*-Hip techniques				*Ashi-waza-Foot*/Leg techniques			
*Ju-no-kata*											
*Ikkyō* (1st group): *Tsukidashi*; *Kata-oshi*; *Ryōte-dori*; *Kata-mawashi*; *Ago-oshi*											
*Seiryoku Zen’yo Kokumin Taiiku*											
*Tandoku-renshu* (solo practice): 1st group—*Kagami-migaki*—2nd group											
*Additional techniques*											
*Tachi-waza* (standing techniques)											
Te-waza: Seoi Nage (ippon/morote/eri); Tai Otoshi											
Koshi-waza: Uki Goshi; O Goshi; Uchi Mata; Harai Goshi; Koshi Guruma; Tsurikomi Goshi											
Ashi-waza: De Ashi Barai; Okuri Ashi Barai; O Soto Gari; O Uchi Gari; Sasae Tsurikomi Ashi; Hiza Guruma; Uchi Mata; Ko Soto Gari; Ko Uchi Gari											
*Ne-waza* (ground techniques)											
Osae-komi-waza (pinning techniques): Kesa Gatame; Yoko Shiho Gatame; Kami Shiho Gatame; Tate Shiho Gatame											
Shime Waza (choking techniques): Kata Juji Jime; Nami Juji Jime; Gyaku Juji Jime; Hadaka Jime; Okuri Eri Jime											
Kansetsu-waza (joint lock techniques): Ude Garami; Ude Ishigi Juji Gatame; Ude Gatame; Waki Gatame											

Note: * FIJLKAM: Italian Judo, Wrestling, Karate, and Martial Arts Federation.

**Table 2 sports-12-00310-t002:** Means and Standard Deviations (SD) of trait anxiety (STAI-Y2); recorded during a rest day and state anxiety (STAI-Y1); Recorded Before (PRE) and at the End (POST) of the simulated and examination conditions.

	STAY-Y2 (pt)		STAI-Y1 (pt)							
	Rest		Simulation				Examination			
			PRE		POST		PRE		POST	
	Tori	Uke	Tori	Uke	Tori	Uke	Tori	Uke	Tori	Uke
** *Mean* **	34.8	35.8	43.2	48.2	45.7	45.8	51.5	55.2	37.4	47.2
** *SD* **	7.8	11.2	6.9	7.6	11.6	8.8	11.1	7.8	9.0	11.1

**Table 3 sports-12-00310-t003:** Means and Standard Deviations (SD) of enjoyment (ENJ), ratings of perceived exertion (RPE), and fear of falling (FoF) were recorded at the end of the simulated and examination conditions.

	FoF (pt)				RPE (pt)				ENJ (pt)			
	Simulation		Examination		Simulation		Examination		Simulation		Examination	
	Tori	Uke	Tori	Uke	Tori	Uke	Tori	Uke	Tori	Uke	Tori	Uke
** *Mean* **	1.7	1.8	1.5	1.3	4.5	6.0	2.7	6.3	9.7	8.8	9.0	7.2
** *SD* **	1.0	0.8	0.8	0.5	2.7	2.4	1.6	2.7	0.8	0.4	0.9	1.7

## Data Availability

The data presented in this study are available on request from the corresponding author.

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
