# Peer review of "Stress-Related Hormonal and Psychological Changes to Simulated and Official Judo Black Belt Examination in Older Tori and Adult Uke: An Exploratory Observational Study"

_sports, 2024, doi:10.3390/sports12110310_

Round 1

Reviewer 1 Report

Comments and Suggestions for Authors

The study contains serious errors in the Materials and Methods and Results sections, which significantly undermine the reliability of the findings. In particular, inconsistencies in timing, data presentation, and measurement procedures prevent the results from being accurately interpreted. Therefore, these major errors must be corrected before the study can be reconsidered.

Wishing you success with the revisions, and best regards.

Kind regards

Abstract

The abstract of the study presents results that contradict the methods outlined in the materials and methods section. This inconsistency negatively affects the validity and reliability of the study. Major revisions are required in the abstract, and the results must be made compatible with the materials and methods. Otherwise, the scientific value of the study will be questioned.

Line 20-21: It has been noted that there is a discrepancy in the timing of the T5 sample between the summary, the Materials and Methods section, and the graphs. The timing of the T5 sample during the recovery phase should be clarified, specifying whether it was taken at the 15th or 60th minute. This inconsistency is crucial for the accurate interpretation of the results, and the necessary corrections should be made in the relevant sections.

Line 23:  In the text, the expression '35.3±9.2Tpt' may cause confusion as the abbreviation 'Tpt' is included within the same parentheses as the numerical values. It is recommended that the 'Tpt' abbreviation be clearly separated from the numerical values in the text. For example, expressing it as '(35.3±9.2) Tpt' would improve readability and clarity."

Introduction

The introduction provides sufficient information, effectively highlighting the purpose and significance of the study. Overall, its contribution to the literature is clearly expressed. However, minor revisions are needed to improve focus and clarity in certain sections. Specifically, the main hypotheses of the study and how they address gaps in the literature could be more explicitly stated. Additionally, some sections could be condensed to avoid unnecessary details and better direct the reader’s attention to the core subject. These recommendations are outlined below.

Line 55: The phrase 'Master athletes are considered examples of successful aging' may typically be used to describe older and experienced athletes, but this term may not be entirely clear to all readers. In an academic text, such terms should be defined more clearly and comprehensively. Additionally, the structure of the sentence in which this phrase appears may complicate the understanding of the text. Therefore, I suggest providing a more detailed explanation of the term 'master athletes' and revising the sentence structure to improve the flow of the text.

Line 68: The sentence beginning with 'Since 1983, the 20-item State-Trait Anxiety Inventories' dedicates too much detail to the STAI, which may negatively affect the flow of the text and the reader’s focus. Instead of providing an extensive explanation of the inventory, a brief and concise description should be given, highlighting why the STAI was used in the study and how this psychological measure relates to physiological stress markers (such as cortisol, testosterone, alpha-amylase). The purpose and use of the inventory should be mentioned, linking the results to physiological aspects to focus on how psychophysiological stress responses are addressed more broadly. Additionally, briefly touching on how physical activity can reduce anxiety will suffice, as the primary focus of the study is on the psychophysiological changes in older judoka. This approach would make the text more coherent and emphasize the main focus of the study more clearly.

Materials and Methods

1.      The experimental design stages are clearly outlined; however, the sample size (N=12) appears small. This limited sample size may reduce the generalizability of the results. Therefore, it is recommended that more information be provided regarding how the sample size was determined and how this size impacts the statistical power of the study. Details such as the criteria used for determining sample size, the targeted effect size, and any prior power analyses should be explained to enhance the validity of the study.

2.      The details of the statistical methods should be explained in greater depth; this should include how the assumptions of the tests used (such as normality and homogeneity of variance) were tested and whether any violations of these assumptions occurred. If any assumptions were violated, the methods used to address this (e.g., non-parametric tests or corrections) should be specified (Line 180-188).

3.      It may be necessary to provide more information about how the simulated examination was organized and under what conditions the participants were evaluated. For example, the similarities and differences between the simulated examination and the actual examination should be specified (Line 86-89).

4.      Defining older Tori participants as aged 65 and above is reasonable. However, considering the physical differences in older individuals, it should be explained how the effects of age will be controlled or taken into account. (Line 94)

5.      The saliva sampling protocol should be explained in detail. To ensure consistency in the sampling conditions, factors such as the time of day when participants were sampled, the temperature, and humidity level of the work environment should be specified. Additionally, the significance of the chosen sampling times (T0, T1, T2, T3, T4, T5) regarding physiological responses should be justified. (Line 169-174)

6.      The process for collecting saliva samples is well explained; however, it is important to provide information regarding the accuracy and reliability of the test kits used for analyzing these samples. To this end, the measurement ranges and sensitivities of the kits should also be specified. (Line 174-178)

7.      The determination of protocol timing is crucial due to the influence of biological rhythms on hormones such as cortisol and testosterone. The levels of these hormones vary according to the time of day; for example, cortisol levels are higher in the morning and lower in the evening. Therefore, it is essential to specify the times at which the measurements were taken. Considering the time factor is a critical element for enhancing the reliability of the study.

8.      The omission of the reliability levels of the psychological questionnaires used (e.g., STAI-Y1, STAI-Y2, CR-10 RPE, FES, VAS) leads to a significant lack of information regarding how consistent and accurate these measures are within the study. Failure to report reliability coefficients (e.g., Cronbach's alpha) may create uncertainty about the reliability of the results. Therefore, providing the reliability coefficients of the questionnaires used is necessary to enhance the robustness and reliability of the study’s findings. (Line 157-167)

9.      It has been observed that there is a discrepancy regarding the timing of the T5 sample between the summary, the Materials and Methods section, and the graphs. Specifically, the timing of the T5 sample during the recovery phase is unclear, with conflicting information suggesting it was taken either at the 15th or 60th minute. This inconsistency is critical for the accurate interpretation of the results. Therefore, it is essential to clarify the exact timing of the T5 sample and make the necessary corrections in all relevant sections of the study to ensure consistency and clarity (Line 113-117).

Results

Line 189: Although testosterone measurements were conducted in the study, these measurements were not presented according to the specific time points, which is a significant omission. Failing to report hormone measurements over time greatly impacts the accuracy and reliability of the findings. Specifically, testosterone measurements were not included in the table and were not presented according to the time points. This makes it difficult to track hormonal responses over time and results in a significant loss of valuable data. Correcting this omission is crucial to enhance the scientific value of the study and ensure the consistency of the results.

Line 218:  There seems to be an error in the description of the sampling times in Figure 1. Specifically, the timing for the T5 sample during the recovery phase is incorrectly listed as "15 min." Since the T5 sample should represent a different time point (possibly the 60th minute), this discrepancy needs to be corrected. The description should accurately reflect the actual timing for all samples to avoid confusion and ensure accurate interpretation of the results. Please revise the timing for T5 in both the figure and the corresponding text.

Line 232: The timing for the T5 sample in Figure 2 is incorrectly listed as "15 min." Please correct it to accurately reflect the actual sample time.

·         Although testosterone (sT) measurements were conducted in the study, they were not presented according to specific time points. The lack of temporal reporting for these hormone measurements makes it difficult to accurately assess changes in testosterone levels over time, reducing the reliability of the findings. Presenting testosterone measurements based on time points in the Results section is crucial to enhance the scientific value of the study and ensure the consistency of the conclusions.

·         There are inconsistencies between the interpretation and numbering of the tables in the study. The references to the tables in the text do not follow the correct numerical order. This issue complicates the understanding of the results and negatively impacts the coherence of the study. Each table should be cited in the text in the order of its first appearance, ensuring that the tables are consistently and properly referenced throughout the manuscript.

Formun Üstü

Discussion & Advantages and shortcomings of the study

After correcting the errors in the Materials and Methods, as well as the Results sections, the Discussion, Conclusion, and "Advantages and Shortcomings of the Study" sections will also need to be rewritten. The inconsistencies and omissions regarding timing, data presentation, and measurement procedures in the Materials and Methods currently prevent the findings from being properly and accurately discussed. Additionally, there are errors and inconsistencies in the Results section, which further undermine the reliability of the conclusions. These issues in both the Materials and Methods and Results sections significantly affect the reliability of the data and make it difficult for the study to stand on solid scientific grounds. Therefore, these sections must be corrected to ensure a better understanding of the study's findings and to enhance its scientific integrity. Once these corrections are made, the advantages and shortcomings of the study can be more accurately assessed, and the conclusions can be rewritten on a more reliable basis.

Author Response

We thank the Reviewer 1 for his/her precious comments. Answers are included in the attached file, reporting all the necessary changes we addressed in the revision process.

Reviewer 2 Report

Comments and Suggestions for Authors

The idea of ​​this study is interesting, my recommendations are the following:

Abstract - I recommend mentioning specifically how the characteristic anxiety was analyzed, the tools used.

Lines 26-27 recommend rewriting, in the sense of concrete mention of the results between the two samples.

Section 2.1. I recommend mentioning the type of study.

Lines 90-98 recommend moving to the Participants section.

Lines 102-117 recommend to be moved, without duplicating the information in sections 2.3.

Section 2.3. Procedures recommend to be divided into two subsections, namely Procedures - where to mention the ways of collecting the results and Instruments or Measurements - where to present concretely, for each aspect concerned - the test instruments. Subsections 2.3.2 and 2.3.3 should be incorporated in the Measurements section.

Lines 119-122 recommend expanding the targeted aspects, mentioning the number of items, the method of selecting the answers.

I recommend that at the end of the Discussion section, the limits of this study, the practical implications and the future research directions should be specifically mentioned.

Lines 350-359 recommend revision, the aspects are presented too generally, unfocused.

Lines 361-363 recommend deleting or rewriting, it is too general.

Lines 363-366 recommend rewriting, the use of the word - magnitude, is not in accordance with this study.

I recommend that the Conclusions be reviewed, focused on results.

Author Response

We thank the Reviewer 2 for his/her precious comments. Answers are included in the attached file, reporting all the necessary changes we addressed in the revision process.

Round 2

Reviewer 1 Report

Comments and Suggestions for Authors

Dear Authors,

Thank you for carefully considering and implementing all the suggested revisions. The changes, particularly in the Materials and Methods and Results sections, have significantly improved the coherence and reliability of the study, making the data interpretation more robust. With these revisions, I believe your manuscript is now of publishable quality and will offer meaningful insights in the field.

Wishing you success in your future research endeavors.

Reviewer 2 Report

Comments and Suggestions for Authors

no comments